# The Poor as Symptom: A Lacanian Reading of the Option for the Poor

Levi Checketts

Department of Religion and Philosophy, Hong Kong Baptist University, Hong Kong SAR, China; checketts@hkbu.edu.hk

**Abstract:** Latin American liberation theology contributed perhaps the most significant theological contribution of the twentieth century in the "preferential option for the poor". This insight has been an uneasy call to conscience for the magisterial Catholic Church, which has often buttressed the positions of the powerful. However, despite the central significance of this discovery, liberation theologians themselves often betray their own positions by romanticizing the poor, speaking on their behalf, diluting the meaning of poor and other such seeming shortcomings. This article argues that the incongruence regnant in discussions of the preferential option can best be understood through the Lacanian notion of a "symptom". As "woman is the symptom of man", the poor are the symptom of the upper classes. In order for nonpoor to understand their own socioeconomic position—including academically trained clergy—they must posit the poor as an Other against whom they understand themselves. As such, reaching "the poor" is an impossibility for anyone who is in a position to truly advocate for them. However, the insight of the preferential option tells us that the impossibility should be pursued nonetheless, with full understanding that it is an impossibility.

**Keywords:** Jacques Lacan; option for the poor; liberation theology; Emmanuel Levinas; ideology





## 1. Introduction

The "preferential option for the poor" has enjoyed being the center of focus in Latin American Catholic liberation theology (hereafter referred to simply as "liberation theology") since it was coined in documents leading up to the Second Latin American Episcopal Conference in Medellin, Colombia. Although liberation theology has remained a controversial mode of theology within mainstream Catholicism, this central tenet was formally adopted into magisterial teaching in Pope John Paul II's social encyclical *Sollicitudo Rei Socialis* in 1987 (John Paul II 1987). While Pope Francis has expressed some criticism of what he considers liberation theology's excesses, some have argued that his papacy has "reconciled" liberation theology with the Vatican (Løland 2021).

It is tempting to say that the mainstream tolerance of liberation theology, as well as its expansion beyond merely the "poor" to include various other marginalized groups (e.g., in various other branches of liberation theology such as black liberation theology, Mujerista theology, Minjung theology and so forth), shows the unquestionable significance of this tradition. However, two challenges remain open. The first is the challenge ushered by more conservative strands of theology. The passing of fifty-five years, the end of the Cold War and the ravages of global capitalism have not been enough to soften the visceral reaction of some against any perceived creep in of Marxist ideology or methodology. This is a point of ideological dispute, so the intractability can only be resolved through ideological means. The other problem, however, is more subtle, and it is the subject of this paper. As liberation theology has gained more prominence it has, ironically, dulled its prophetic bite.

Liberation theology's preferential option has always been its strength and is perhaps the single greatest theological insight of the twentieth century (cf. Goizueta 2003, p. 143). This insight, liberation theologians are right to note, requires an entire reorientation of

our life—a metanoiac transformation. The simple truth of this insight has led to the prominence of liberation theology across the Catholic world. However, when this statement is expanded, liberation theologians often contradict themselves. The poor must be listened to, for example, but are said to be voiceless. Or the poor are our guarantors of salvation, but on the other hand, they themselves have to be evangelized. This seeming contradiction is under-explored in liberation theology. How can we make sense of this problem in good faith? I contend that Lacanian psychoanalysis gives us critical insight into this problem. The poor, as "the poor", exist as "symptom" of the upper classes, including liberation theologians. Because of the structure of our symbolic orders, it is impossible to simply break out of this framework, but the Christian eschatological orientation and a Levinasian "unsaying" help us recall that our own psychological limitations require us to place our hope in the perfecting work of God who alone is able to overcome the "impossibility".

## 2. A New Word: The Preferential Option

It is no great secret that Christianity writ large and Catholicism in a special way have too often been intertwined with the interests of the most powerful in society. Whether one looks at Constantine convening the Council of Nicaea, the conflict of crown and cross in the Investiture Controversy, the elision of military and religious aims in the Crusades, the secret policing of the Spanish Inquisition, the papal blessing given to the conquistadors or the ultra-monism of the First Vatican Council as the Italian Army took Rome, it is undeniable that throughout Catholicism's history, the hierarchy has long had a vested (or invested) interest in questions of political, military and economic power. Indeed, this is so typical that Karl Marx declared in "On the Jewish Question" that a state that claims to be Christian "can only escape its inner torment by becoming the *myrmidon* of the Catholic Church" (Marx 1978, p. 38, emphasis original). Thus, Marx's own atheism is rooted in the assumption that religion is inherently a tool of hegemony, and that a classless society would have to be rid of the power structure of religion. For roughly one hundred years, the magisterial Catholic Church maintained the corollary to this view—Marxism must be inherently anti-Christian and so must be opposed in its entirety.

At the same time, Catholicism itself seemed hard-pressed to prove Marx correct by buttressing the powerful and elite. Take the situation in the United States, for example. While early Catholic immigrants often were themselves marginalized and relegated to second class citizens, American Catholic leaders such as Fulton Sheen and Isaac Hecker wanted to convince the largely Protestant country that Catholicism was not only compatible with American mores but was the embodiment of the American spirit. Thus, despite anti-Catholic worries that an American president would be a vassal to Rome, the U.S. elected their first Catholic president in 1960. Few American Catholics seemed to think this was unfortunate—only radicals of the sort such as Dorothy Day, who was herself the frequent subject of ecclesial censure and reprimands, would deny that Catholics should hope to be fully integrated into American society. On a larger scale, the magisterium seemed reluctant to critique the machinery of power in the Industrial age. Pope Leo XIII's *Rerum Novarum*, to take the prototypical example, does critique abuses of power by industrialists, but places an equal burden of responsibility on the worker to cooperate with their oppressor (Leo XIII 1891). Similarly, Pius XI's *Quadregesimo Anno* asks capitalists to be more humane in their capitalism but flat out condemns communism as being without any hope of redemption (Pius XI 1931, para. 112).

The 1960s presented opportunities for a change in perceptions and understanding of the poor and the meaning of Marxism in Catholic theology. In 1963, following the Cuban Missile Crisis, when the U.S. nearly engaged in a nuclear war with the Soviet Union, Pope John XXIII issued the encyclical *Pacem in Terris*, which affirms, among other things, the state's responsibility to ensure basic economic rights for its citizens (John XXIII 1963, para. 65). At the same time, the Second Vatican Council was discussing new challenges to the Church, which would manifest in three distinct stance changes that opened the door for liberation theology. First, the opening line of the Pastoral Constitution

for the Church in the Modern World prioritizes the poor: "The joys and the hopes, the griefs and the anxieties of the men of this age, *especially those who are poor* or in any way afflicted, these are the joys and hopes, the griefs and anxieties of the followers of Christ" (Paul VI 1965a, para. 1, emphasis added). The Church's rhetorical commitment to align itself with the interests of the poor may not have been borne out fully in most countries following the council, but it gave justification for those who would take this position seriously. Second, the same document emphasizes the need of theologians to use new methodologies to help the modern age understand the deposit of faith (Paul VI 1965a, para. 62). Finally, the Pastoral Decree on the Office of the Bishops emphasizes the importance of local bishops' councils to address the spiritual and moral needs of their own populations (Paul VI 1965b, para. 38). Taken together, these three important changes open the gates for local theologians, under the guidance of their local bishops, to use those methodologies that best express "the joys and the hopes, the griefs and the anxieties" of the poor as the proper task of theology. Thus, three years later, the Second Latin American Bishops Episcopal Conference gathered in Medellin, Colombia, where the phrase "preferential option for the poor" was first recognized as a central tenet of their position.

The emphases on new methodologies and the experience of the poor meant, to theologians such as Gustavo Gutierrez, Leonardo Boff, Juan Luis Segundo and others, that the Marxian notion of praxis—a process of taking one's assumptions of the world, trying to enact change, observing the outcomes, reflecting on the meaning and repeating—was the best option for directing the faith to the interests of the worst off. The tri-fold methodology of "look, judge, act", responding to social ills outlined by John XXIII in *Mater et Magistra* (and previously developed by Joseph Cardijn) (John XXIII 1961, para. 236), aligned well with Marxian praxis, at least as a methodology. And so Latin American clergy began reflecting on the lived poverty of the faithful in their communities, the origins of these problems and the needs of their church. In the Second Conference of the Latin American Bishops, the poverty of the region and its pervasiveness was a central theme, and the bishops called for the Church to be a "poor church" directed to "a sign and a commitment—a sign of the inestimable value of the poor in the eyes of God, an obligation of solidarity with those who suffer" (CELAM 1968, para. 7). From this cue, the first generation of liberation theologians took the orientation of poverty as a central commitment in their theological focus.

The expression "preferential option for the poor" is the most important symbol of this development of thought. Depending on which theologian one is reading, the expression means a prioritization of human rights (Boff 1989, p. 14), a "locus theologicus" of God "preferentially identified with the victims of history" (Goizueta 2003, p. 144), the "salvation and liberation" of those through whom "the mystery of reality breaks through" (Sobrino 2008, p. 19) or "solidarity with the poor and protest against the inhuman situation of poverty" (Gutierrez 2007, p. 30). The variations of interpretations suggest that the phrase itself signifies more than a simple dogmatic statement. But chiefly, the expression emphasizes the priority of the poor above the interests of the wealthy. The tacit or express cooperation of the Church with the elite is explicitly denounced. The "church of the poor" is oriented toward the salvation and interests of the worst off. The theology that articulates the experience of poverty is not the theology that the powerful have wielded. The Christ of liberation theology is a Jesus who truly dwelt among us, a God who suffers with the suffering.

Liberation theology gained notoriety, however, not because of this prioritization of the poor but because, in adopting a perspective of poverty, they recognized the conflictual nature of society. Marx's insistence that society is contextualized by conflict is not well regarded by churchmen who have interests tied to the status quo. Paulo Freire's language of "oppression" entails a clear demarcation of victims and perpetrators in society, and if the church has not always adopted the perspective of the poor, they must have propped up the guilty (Freire 2017). The language of class consciousness, so often tied to language supporting violent revolution, has been regarded as suspect, and liberation theologians have been regarded as potential instigators. Nonetheless, in a true imitation of Christ, liberation theologians insist on peace and harmony, all while the poor are devoured by the

rich. It is, in fact, not the liberation theologians who support violence, but the institutional church which props up systems of violence and oppression.

Liberation theology has earned the ire of many comfortable Catholics perhaps most sharply because it holds a mirror up to them. Thus, although Josef Ratzinger (Pope Benedict XVI) condemns the violent injustice resultant from centuries of capitalist exploitation, he more strongly condemns the violence inherent in Marxist thought (Ratzinger 1984, sct. VIII). Just as Pope Leo XIII a century before, Ratzinger gives a pass to the violence of the powerful by placing the burden of peace on the weak. However, the witness of liberation theologians is vindicated in the blood of their martyrdom, those whose deaths disprove the inflammatory rhetoric. Famous examples, of course, include the assassination of San Salvador Archbishop Oscar Romero and the murder of the six Jesuit professors at the University of Central America. Liberation theology sees this cost as worthwhile because it is the cost the poor pay. As Jon Sobrino notes, the poor are those who are unable to take their lives for granted (Sobrino 2008, p. 58). They are the nameless "millions of victims and martyrs" of violence, and their witness demands our response (ibid., p. 17). The witness of liberation theologians, some of whom testified with their own blood (see Boff 1989, p. 11; Sobrino 2007), stands strong against the claims of an indifferent society and a church that has too often been silent on the issue of oppression.

### 3. Contradictions and Impasses

The results of liberation theologies across different cultures and perspectives should not be downplayed. But the preferential option articulated by Latin American liberation theologians has significant shortcomings, namely that it rarely accomplishes its goal of articulating theology from the perspective of the poor. Indeed, the theologies of liberation theologians often contain contradictions or blind spots when trying to lift up the voices of the so-called voiceless. There are a number of ways these shortcomings appear, but all of them to one degree or another are indicative of a hermeneutical impasse, a failure to truly understand the perspective they are championing. This failure, I contend, owes to the cultural background of liberation theologians. The poor are, to them, "symptom" in a Lacanian sense—a defining part of their identity existing outside of them. This not only explains the apparent impasses but also their inability to notice the impasses.

One of the more obvious failures is the notion of being "voices for the voiceless". In numerous expressions of liberation theology, the poor are characterized as the "nameless" and the "voiceless"—those who lack status in society and those who lack power. As Gustavo Gutierrez points out, we are to listen to the poor and "not pretend to be—as is said many times with goodwill that we are all aware of—'the voice of the voiceless'" (Gutierrez 2007, p. 31). And yet, his nuanced caution reveals that somehow liberation theology has acquired the reputation of being a voice for the voiceless. This is not in the least surprising, however, as Gutierrez himself, only a few pages later, takes it upon himself to articulate the perspective and to be the "voice" of the "nameless" woman in the Gospel of Mark who washes Christ's feet (ibid., 34). Similar parallel contradictions persist across liberation theology writings. The poor are those who reveal to us our own sinfulness (Sobrino 2008, p. 49), but at the same time they must be conscientized through evangelization (Gutierrez 1973, p. 116). The poor are the clients of Yahweh who have special privilege before God (Gutierrez 1973, p. 296) but are also the "crucified people" who appear to us as though abandoned by God (Sobrino 2008, pp. 4–5). The poor are the "nameless thousands" (Sobrino 2007, p. 99; cf. Hartnett 2003), while liberation theologians are named, even by each other, as models of faith (see: Sobrino 2007, pp. 99–100, 101). Ultimately, this impasse reveals the problem that the "voicelessness" of the poor allows them to stand in as proxies for the liberation theologians themselves. In Gayatri Spivak's provocative phrase, "the ventriloquism of the subaltern is the left intellectual's stock-in-trade" (Spivak 2010, p. 27); the "voicelessness" and "namelessness" of the poor is not the brute silence that defies our comfortable moral standing but rather serves as the bona fides for the speaking liberation theologian.

A second failure, less pervasive perhaps than the first, is the notion, in Stephen Bede Scharper's words, that "in order to opt for the poor, one must be nonpoor" (Scharper 2014, p. 99). "With goodwill we are all aware of" (to borrow Gutierrez's phrase again) as this notion is, it is quite obviously untrue. In the first place, this notion contradicts the central virtue of solidarity in liberation theology. How can one be in solidarity with the poor if one claims that the poor cannot advocate on their own behalf? How can the poor participate in their own liberation (Gutierrez 1973, p. 113)? The poor can and do advocate for themselves, and even, as noted by numerous Marxists, for the rich through the process of "false consciousness". Indeed, Freire's notion of conscientization, important in Gutierrez's theology of liberation (Gutierrez 1973, p. 116; cf. Freire 2017, p. 41), insists that the poor can and should be active in advocating for themselves; though, once again, this assumes that the non-poor contain the correct information, which must be imparted to the misinformed poor. Thus, the larger failure that Scharper's unfortunate phrase reveals is the assumption of the "entitled advocate" (Taylor 2003). Liberation theologians, themselves often coming from upper-middle class backgrounds and wealthier Western nations with pedigreed educational backgrounds and all of the due trappings for living among the professional class (if only as pariahs), write as and to "people of privilege" (Groody and Gutierrez 2014, p. 6). Their assumed position is the upper classes and their assumed readers are of the same category. Scharper's expression reveals, then, that much of the conversation of liberation theology has been *by* and *to* the elite *about* the poor. As such, the preferential option for the poor has often meant the objectification of the poor, those "nameless thousands" who are talked about and not talked with.

In a third "goodwill" failure, the notion of "option for the poor" was undermined over time by the evolution of the notion of "poor and marginalized". Initially, this movement began as a recognition, in line with Kimberle Crenshaw's insights on "intersectionality", that poverty is exacerbated by other forms of oppression (Crenshaw 1993). Thus, theologians lifted up in an important way the concerns of U.S. Latinos and African Americans (Elizondo 2007; Copeland 2007), women (Aquino 2007; Hilkert 2007; Tamez 2007), LGBTQ persons (Tamez 2007) and peoples in Asia and Africa (Kalilombe 2007; Pieris 2007). But this shift also allowed for a sort of duplicitous movement. Virgilio Elizondo, for example, notes: "No matter how much individuals of [minority] ethnic backgrounds succeed *economically*, educationally and professionally, the dominant culture by and large still considered them inferior others" (Elizondo 2007, p. 160, emphasis mine). As he points out at the beginning of his essay, despite ascending into the professional class, Elizondo still felt disfavored by virtue of his ethnic background as a Mexican American. The *despite* is telling: Elizondo willingly flees poverty but cannot flee his ethnicity. Thus, he adapts the notion of "poor" to mean something beyond mere material poverty (cf. Hartnett 2003) to include the "spiritually" and "existentially" poor, some of whom are in the upper socioeconomic classes (Elizondo 2007, p. 159). When option for the poor and marginalized eventually means even options for those who exploit the poor, then the symbol has become entirely distorted.

A fourth contradiction lies in the romanticization of the poor. Jon Sobrino cautions against this tendency, recognizing the inhumanity of the condition of created poverty (Sobrino 2008, p. 9). And yet, Sobrino himself speaks quite romantically about the poor—the "crucified people", those who bring us salvation (ibid., 3, p. 49). Other liberation theologians repeat this error, waxing lyrically about the "the just ones" (Gutierrez 1973, p. 297), the happy poor (Sobrino 2008, p. 15), spiritual children (Gutierrez 1973, p. 297; Boff 1989, p. 23), icons (Sobrino 2008, p. 75), those who hear the divine call (Boff 1989, p. 81), God's "suffering servant" (Gutierrez 1973, p. 202; Boff 1989, p. 86; Sobrino 2008, p. 4) and so forth. The overall image one gets from this view is that the poor are entirely innocent saints. While this aims to correct the vilification of the poor (a term that itself bears the traces of anti-poor sentiment), the result is the poor remain useful abstractions and not real persons. One should understand the poor as sinners and saints, as holy and vile, as persons who do make decisions—constrained as they may be—within a constellation of options. To describe the poor as mere victims, or to impinge on them the burden of granting our salvation to us, is

to reinscribe the process of erasing their subjectivity. It is to continue to fail to see the faces or hear the voices of the poor as they are relegated to types.

Fifth, then, this leads to the question of why, after sixty years of liberation theology, are the poor still silent? Why have we not heard the poor in their own words? Gayatri Spivak's provocative question "can the subaltern speak?" seems to be answered negatively by liberation theologians who describe the poor as voiceless. Indeed, the reality is rather that theologians generally—including liberation theologians—have not learned to listen to the poor. To say the poor are voiceless is another way of saying they are unheard. The poor do, in fact, speak. Indeed, the poor quite often have to speak in the language that the upper classes will understand. Consider "Mrs. H", a poor Irish mother pleading to the Archbishop of Dublin for aid. She knows the archbishop is perhaps more concerned about losing Catholic faithful to Protestants than about the deaths of the poor (assuming, perhaps, that the faithful poor will wind up in heaven). Thus, to incentivize the archbishop to hastily send aid, she threatens apostasy, noting that a relative has promised aid if she becomes a Protestant, though she has resisted so far (Earner-Byrne 2017, pp. 14–15). Irish historian Lindsey Earner-Byrne notes several other such "speech" acts of the poor who understand how they are seen by the upper classes—they characterize themselves as pious, down on their luck, victims of circumstance, repentant sinners or potential apostates, all to elicit sympathy to their cause.

Thus, a final impasse: the upper classes must educate the poor for their liberation while claiming the poor are voiceless. In fact, the poor speak, but are not heard because pedagogy itself is a type of master discourse. Thus arises the apparent irony that liberation theologians are to, on one hand, adopt the perspective of the worst off, but also, on the other, educate the oppressed to their oppression. The process of praxis, meant to be transformative and dynamic, is a feedback loop. The assumption that the poor are oppressed fuels the adopted perspective of poverty, namely the view that the poor are oppressed and silenced. The "conscientized" upper classes then educate the poor to conscientize them of their own oppression. If the poor accept this, then the assumption that the poor were hapless victims unable to speak their own perspective is verified. If the poor reject this, they are deemed victims of false consciousness. It is important to note that whether or not the assumption that the poor are oppressed is true is irrelevant; what matters in this situation is that the so-called option for the poor actually begins as an option from the perspective of the non-poor and is ascribed to the poor after the fact.

The beginning of a way out of these impasses would be to start a liberation theology that begins authentically from the voice of the poor. This is a long-standing challenge of Marxism generally—the revolution of the proletariat should actually be led by the proletariat and not by intellectual or military leaders, as has typically been the case. However, the degree to which the poor accept the proposition that they are oppressed and need to overturn the system of their oppression varies from place to place. The poor understand their position as poor in various forms—as misfortune, as fate, as cosmic justice, as a temporary setback or as truly human-inflicted injustice. The meaning of their poverty, as well as whether or how they wish to escape it, will depend in part on their location in culture, economic systems, political structures and religious beliefs. As a result, it is not unfair for liberation theologians to declare that the poor are oppressed—they need only look at economic history and its social ramifications—but to say this as though they were speaking from the voice of the poor is disingenuous. To declare that the poor need to be liberated and that one is going to help the poor in understanding their need for liberation is fine as long as one is willing to confess that it is a declaration made from above and not from below. But to avoid the temptation to speak on behalf of the poor, one must be ready to "unsay" the declaration that the poor need liberation even as it is proclaimed.

## 4. The Poor as Symptom

This leads to a greater problem, however, which is illustrated by these impasses, namely that there are no "the poor". Awkward as this expression is, it reminds us that

the "nameless" poor remain nameless in part because they are constructed as a monolithic entity. They are not Rosita Herrera or Jose Lopez, peasants living in a shack with four children and a handful of chickens on a farm they work for a landlord—they are "the poor". "The poor" must remain nameless because they are an abstraction, a construction of the bourgeoisie necessary to create the class unity of the dominant class. As a monolithic entity, the poor are, in the words of Jacques Lacan, the "symptom" of the upper classes. There are, of course, poor people, individuals with life histories, cultural backgrounds, networks of relations and so forth—specific Johns and Marys, Josés and Marias, Seans and Mollys, Cheol-sus and Young-hees—but no "the poor".

The intimations of this fact are traced out through Liberation Theology itself. Marcella Althaus-Reid, for example, notes that "the poor" described by first-generation Catholic liberation theologians are not the "urban poor" of Latin America. Her work explores sex and gender as a lens for talking about "the poor", including the prostitutes of Buenos Aires. These "the poor" are different from those "the poor" whom Gutierrez and Sobrino write about. "The poor" of post-Medellin Liberation Theology exist conceptually as Latin American peasants, and thus, other groups of the poor were left out. Often enough "the poor" are spoken of as entire nations, or even, as Gutierrez is wont to do, even entire continents. (Nonetheless, we must not look too closely at even this designation, lest we question whether Latin American middle classes should be included while North American homeless are not.) "The poor" of liberation theology are voiceless and anonymous because they are a monolithic entity. How can an entire peasant population speak with one voice? What name can the throngs of a continent have?

The non-existence of the poor is demonstrated, ironically, by the well-intentioned writings of those advocating for them. Where the liberation theologians say "the poor", one can read the same sentiments, ideals and apparently class spirit in Spivak's "subaltern", Marx's "proletariat" or Foucault's "plebs" (Foucault 1980). These are the masses Spivak so wryly notes function as puppets for left intellectuals. So, unsurprisingly, the interests of Foucault's plebs are his interests, as are those of Marx's proletarians. Indeed, the insistence upon the "rightness" of the perspective of one's projected alter-class is so sharply delineated that Lukacs can claim, on one hand, that only the proletariat can experience class consciousness because "the outlook of the [peasants] is ambiguous or sterile because their existence is not based exclusively on their role in the capitalist system of production" (Lukacs 2017, p. 47), while Fanon argues "only the peasantry is revolutionary. It has nothing to lose and everything to gain" (Fanon 2021, p. 22). The dialectically opposed readings of the peasantry and the proletariat between these two critical thinkers demonstrate the disconnect most clearly: the poor, whom they envision as being most in line with their own philosophy, are meant to be the ones who embody the Hegelian World Spirit that expresses itself in the act of revolution.

The reason Catholic theology still talks about an option for "the poor" (with or without the added "and oppressed") is because "the poor" is a necessary fiction for bourgeois theology. They are "symptom" for the bourgeoisie as "woman is a symptom" for man (Lacan 1975, p. 65). A symptom is "the effect of the symbolic on the Real" (ibid., p. 20). The symbolic constitutes, for Lacan, our entire framework of language and (conscious) understanding of the world. The Real, on the other hand, is all of reality, not all of which fits into our symbolic frameworks. The symptom exists in the liminal space where the symbolic comes up against the Real. The symptom is something that a person "believes in" insofar as the symptom gives meaning to the person's consciousness (ibid., p. 63). Therefore, as "woman does not exist" (ibid., p. 70), "the poor" does not exist. Obviously there are human beings we call women, but to discuss what a woman is is to accept the counterfactual, i.e., that there is man, and, in line with de Beauvoir's insight, woman is defined in relation to this man (de Beauvoir 2010, p. 283). Lacan takes this further—not only is woman defined in relation to man, but the definition of woman constitutes man. Without positing woman, there is no identifier by which man—the dominant gender in patriarchal society—can identify himself. The key, then, is that "the poor" is a symptom of the bourgeoisie (or,

more accurately, the dominant class). "The poor" is the necessary condition for positing a bourgeoisie and especially to establish the bourgeoisie's dominance. This *economic* marker is necessary as a descriptor for all, an umbrella under which the most diverse groups of people (peasants, factory workers, farmhands, panhandlers, service workers, sex workers, thieves, drug dealers, scavengers, cleaners, domestic helpers, nannies and so forth) fall so that the bourgeoisie can recognize *themselves* as one unified group.

Understanding "the poor" as symptom helps illustrate why Marx's hope for the imminent proletarian revolution has never been realized. What Marx and his followers failed to understand is that, despite the "goodwill that we are all aware of", he still approached the question of revolution from the background of the scholar class. For all the discussion of the poor needing to be part of their own liberation, the intelligentsia are more than happy to be the ones leading this charge. It is the educated, those already inhabiting privileged spaces within the dominant classes, who "speak" of "the poor". This unfortunately means that the poor themselves only recognize their classed role if they can see it through the eyes of the fearful bourgeoisie—they can only be radicalized if they see the literate, economically-motivated middle classes as their natural enemy against whom they must unite. While this narrative certainly persists among the bourgeois literati, it beggars belief that the "voiceless" poor would have this same consciousness. Thus, in direct contrast to Lukacs, it is *not* "the proletariat" who alone can have true class consciousness because "the proletariat" are a symptom of the bourgeoisie.

Thus, like the woman who must maintain consciousness of the man, who defines himself by defining her, the poor must be always aware of how they are perceived and defined by the upper classes within the anxious exchange of interpersonal connection (Lacan 2014, p. 264). The example of Mrs. H again proves illustrative: in pleading to the archbishop, she betrays amazing guile and tact—she does not fit the "holy" image of the poor as one who grants salvation—she threatens apostasy if her needs are not met. Earner-Byrne notes many such cases of the poor employing tact, strategy and cunning. These poor appeal to their religious backgrounds, the same backgrounds as liberation theologians' "the poor", but they recognize the intricacies of social, economic and religious dynamics at play. They "know the score" enough to understand how to appeal to fantasies about the "good" poor. But they are only able to do this because they are defined in their position as "the poor" by those whose identity depends on them being a monolithic other to preserve their own unified class distinction.

The reason why Groody and Gutierrez refer to themselves and their audience as "people of privilege" and Scharper claims only the nonpoor can opt for the poor is because "the poor" are a necessary symptom for the "nonpoor" in their identity as "nonpoor", and thus, only in this dialectical construction can "the poor" be advocated for. Thus also, "the poor" can be transmogrified into any number of alternative constellations that function the same way—proletarians, peasants, plebs and so forth, each of which descriptors functions as a stand-in for a general notion of "the poor" against dominant interest groups. To identify concrete individuals within this constellation is to reveal that the construction itself is only possible with its own self-deceptive consciousness. The poor must remain nameless because "the poor" are nameless; any named poor person escapes the tautological constraints of this construction against which the dominant classes define themselves. To be "Joe the plumber" is not to be "the poor" in this mute fashion. In turn, the definition of poverty is subject to redefinition at the whims of the upper classes. As Javier Echeverria notes, the definition of poverty has experienced alterations and redefinitions according to governmental and social interests—whom we want to include as "the poor", whether to expand our own circle of elites or restrict it, to enlarge our social responsibilities or to limit them, is entirely the purview of the upper classes (Echeverria 2014, p. 50).

Because "the poor" remain a symptom of the dominant classes, the poor as constituted in their individuality cannot be aggregated as an anomalous group. Those qualities that liberation theologians or various "people of privilege" attribute to them, whatever their significance for the dominant classes, are true of individuals as often as not. The meek, the

holy, the generous, the honest, the wicked, the naïve, the vicious, the criminal, the lazy, the earnest—every attribute ascribed to invoke a certain sense of who "the poor" are and how they should be treated—are more symptomatic of the person expressing the view than they could ever be to describe people whose sole unifying feature is a lack of material resources. They tell us what we think about ourselves and our relation to the barred Other of "the poor". They reveal our own desires and anxieties. In the case of liberation theologians, "the poor" are often ascribed too many romantic qualities—simple faith, grantors of salvation, innocence and so forth. All those holy attributes we wish to uphold are found in the poor of liberation theology. Recall Sobrino's thesis that martyrdom, the ultimate demonstration of one's Christian faith, is "the maximum expression" of "the reality of the poor" (Sobrino 2007, p. 93). The ideal Christian icon is pasted onto faces constituted by "voicelessness, and anonymity that millions of human beings have suffered" (Sobrino 2008, p. 24). "The poor" stand as an empty signifier onto which we project our innermost anxieties, desires, yearnings and fears.

## 5. Conclusions: Eschatological Tension

One must read the writings of liberation theologians—as is also true for subaltern scholars—with an eye askant. The intellectual pedigrees these scholars represent betray their "epistemology from below". Spivak, a Brahman, studied under no one less than Jacques Derrida and teaches at Columbia University; she occupies one of the most privileged positions a thinker possibly can. It is hard to not read in her own work the "ventriloquism of the left intellectual" she so astutely critiques in others. We see this as well in theologians such as Gutierrez, who studied with Henri de Lubac, Yves Congar and Marie-Dominique Chenu at Leuven, or Igancio Ellacuria and Leonardo Boff who both studied under Karl Rahner. One could hardly call these titans of theology (or philosophy, for Spivak) marginalized voices. Their educational pedigrees are unquestionable, except insofar as they are meant to represent any voices outside of the center.

The aim of liberation theologians to speak from the margins is further complicated by the celebration surrounding their work. After decades of teaching at prestigious institutions such as the University of Notre Dame and Boston College, first-generation liberation theologians' students and these students' students have secured academic pedigrees. The highly competitive nature of these graduate programs ensures that those who succeed their academic seniors are those who are already conformed to the writing and thinking style typical of bourgeois theology, just written in the key of liberation. To the degree that generations of theologians since the 1960s have furthered this work, the revolutionary insight of the "option for the poor" has become a cliché, in some ways a mere designator signifying one's bona fides in the right circles of Catholic theology. The marginalized view of the poor has become a central perspective, made clear by no one less than the head of the Roman Catholic Church, Pope Francis, whose liberation theological engagement and context have given liberation theology renewed interest.

In spite of the way liberation theology has become banal, we should not underestimate the importance of the option for the poor. The freshness of this view can only be maintained, however, through something like the Levinasian concept of "unsaying". In addressing the question of transcendence, that is, what is "otherwise than being", Emmanuel Levinas insists that this can only be "stated in a saying that must also be unsaid in order to thus extract the *otherwise than being* from the said in which it already comes to signify but a *being otherwise*" (Levinas 1998, p. 7, emphasis original). Put in Lacanian terms, we recognize that in the saying of a new term, it enters into the currency of the symbolic order. It enters initially as an irruption, creating a crack or break in the world that exists, revealing the Real beyond (Lacan 1997). The initial iteration of the option for the poor was so significant because it was just such an irruption of the real into the symbolic order. But because subjects cannot exist outside of the symbolic order, this term must be adopted into the symbolic. Just as Heidegger's laborious terms like *Dasein* and *In-der-Welt-sein* became domesticated

by other philosophers, the option for the poor loses its bite because it is adopted as an official position within magisterial theology!

Unsaying the said means to renounce (or denounce) the development of liberation theology as it is being unfurled. This is necessary insofar as liberation theology seeks to speak from the margins, but to speak from the margins means one must never occupy the position of the center. At the same time, liberation theology hopes to be prophetic, to call the powerful to attention. The tension inherent in this work requires continual renunciation. The liberation prophet must, on one hand, remind the "people of privilege" she writes to that the focus is to be the poor and not her, and, on the other hand, she must continually question her own assumptions through the process of praxis, recognizing that the Other who is "the poor" is a barred subject, one whom we can never approach in fullness (Lacan 1998, p. 81; cf. Derrida 2008, p. 82).

In unsaying the said, we recognize that all of the statements made about the poor must be understood only through negation. A proper unsaying in liberation theology involves dialectically opposing those statements which too easily are accepted by the audience to whom the liberation theologian speaks. Thus, instead of Sobrino's "the poor bring salvation", we must say, "the poor testify to our guilt". Instead of "the poor are voiceless", we recognize "the bourgeoisie are deaf to their speech". Instead of "the poor are the crucified people", a statement that sanctifies the senseless evil done to the poor, we must insist "we are the hangmen". Against "the poor must be conscientized", the bourgeois intelligentsia must "unlearn what we are teaching". Most of all, in the unsaying, we must acknowledge that all statements are bourgeois distortions because the process of "legitimate" knowledge production is inherently hegemonic; to be able to speak to the rich means being unable to understand the poor.

Finally, we might unsay the "option for the poor" in the words of Jesus: "The poor you always have with you". Understanding the impossibility of crossing the barred subject to understand what it means to be marginalized, we recognize that the seeming dismissal of the poor by Christ can be understood as a mantra to contextualize our eschatological orientation. In the fallen world we inhabit, the poor we always have with us. No society has succeeded in removing the margins; to remove the margins would be to fulfill the coming Kingdom of God. However, because we are not God, and because we are unable to cross the threshold of the Other, we must recognize that ultimately our action is inadequate. The task of liberation theology, then, is to continually seek the margins, to retreat ever outward. The task is impossible because the poor we always have with us, and so our statements must *always* be understood as inadequate. The saying must always be unsaid. Therefore, the option for the poor is the unsaying of the bourgeoisie, the expanding of the circle, the continual epistemological humility in the face of the other whom we can never fully know because the other is wholly other.

**Funding:** The cost for funding this open-access article was in part sponsored by Hong Kong Baptist University's Research Committee and the Department of Religion and Philosophy.

**Institutional Review Board Statement:** Not applicable.

**Informed Consent Statement:** Not applicable.

**Data Availability Statement:** Not applicable.

**Conflicts of Interest:** The author declares no conflict of interest.

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
