# Peer review of "The Poor as Symptom: A Lacanian Reading of the Option for the Poor"

_religions, doi:10.3390/rel14050639_

Round 1

Reviewer 1 Report

This is an excellent piece of academic writing. Rooted in sound research and theological reflection, it brings a fresh perspective and makes a valid contribution to the existing scholarship.

Though it certainly is publishable without problems, I believe it can nevertheless be improved further in line with the following remarks/suggestions:

1.) Though the main argument of the article corresponds well to its title (the poor as symptom), the actual discussion of the topic of "symptom" in the Lacanian perspective does not take as much space as the reader might expect (lines 350 to 428). This reader at least would be keen to hear more on this concept and its theological reflection.

2.) Similarly, it could be argued that the Levinasian notion of "unsaying" as a potentially important concept is also underdiscussed and, moreover, introduced very late in the article (in the conclusion!). I suggest moving it to one of the main sections of the article and dedicating more space to it.

3.) Furthermore on the concept of "unsaying" and "negation," I believe that some of the examples of negation statements should be reconsidered because of their possibly undesirable side-effect significance. In particular, the claim that "the poor usher in our damnation" (line 483) might evoke an impression that they possess some kind of a demonic quality.

4.) Finally, further consideration should also be given to the reflection on the biblical dictum that "the poor you always have with you." Though the author´s insights are undoubtedly relevant, there is also an inherent danger that these words become a paralyzing factor, leading people to idleness and resignation.  

Author Response

Thank you for your suggestions. I will introduce the notion of symptom earlier and give fuller explanation to it in the section on symptom. I can introduce the notion of unsaying earlier, but I do not think I can give much attention to it earlier without entirely restructuring the paper.

I take your note about the unsaid statements and will adjust the particular one in question to rather say "the poor testify to our guilt."

I realize people may read "The poor you always have with you" as paralyzing. This leads to an entirely other problem, one which I do not have space to address in the paper, namely the apparent conflict between utopian aims and the realities of moral action in the world. The paper is already at word count limit, and even the small changes above may be too much for it without cutting other parts out.

Reviewer 2 Report

The author/authorss of the article chose to advocate an interesting topic of liberation theology and the idea of poor from an interdisciplinary point of view (history, psychology, light-sociology and very little theology itself). Even though the structure of the article is quite clear, the amount of information and formulation of the sentences makes it (at some points) difficult to grasp the full meaning of what the author/authors want to say. It also may have to do with some occasional grammatical and syntactical mistakes - I recommend to submit the article to the English language correction. 

Even though the structure of the article is quite clear, the amount of information and formulation of the sentences makes it (at some points) difficult to grasp the full meaning of what the author/authors want to say. It also may have to do with some occasional grammatical and syntactical mistakes - I recommend to submit the article to the English language correction.

Author Response

I appreciate your feedback. I will review once again my manuscript to see where the grammar is confusing.